# Prediction of Carotid In-Stent Restenosis by Computed Tomography Angiography Carotid Plaque-Based Radiomics

**DOI:** 10.3390/jcm11113234

**Published:** 2022-06-06

**Authors:** Xiaoqing Cheng, Zheng Dong, Jia Liu, Hongxia Li, Changsheng Zhou, Fandong Zhang, Churan Wang, Zhiqiang Zhang, Guangming Lu

**Affiliations:** 1Department of Diagnostic Radiology, Jinling Hospital, Nanjing University School of Medicine, Nanjing 210029, China; rabbitkiller80@126.com (X.C.); jialiu2022@126.com (J.L.); zhouyisheng@hotmail.com (C.Z.); 2Department of Diagnostic Radiology, Xuzhou Medical University, Xuzhou 221004, China; dongzheng96@126.com; 3Department of Diagnostic Radiology, Jinling Hospital, The First School of Clinical Medicine, Southern Medical University, Nanjing 210000, China; sunsetlhx@163.com; 4DeepWise AI Lab, Beijing 100080, China; zhangfandong@deepwise.com (F.Z.); churanwang@pku.edu.cn (C.W.)

**Keywords:** computed tomography angiography, radiomics, carotid artery stenting, restenosis, plaque

## Abstract

In-stent restenosis (ISR) after carotid artery stenting (CAS) critically influences long-term CAS benefits and safety. The study was aimed at screening preoperative ISR-predictive features and developing predictive models. Thus, we retrospectively analyzed clinical and imaging data of 221 patients who underwent pre-CAS carotid computed tomography angiography (CTA) and whose digital subtraction angiography data for verifying ISR presence were available. Carotid plaque characteristics determined using CTA were used to build a traditional model. Backward elimination (likelihood ratio) was used for the radiomics model. Furthermore, a combined model was built using the traditional and radiomics features. Five-fold cross-validation was used to evaluate the accuracy of the trained classifier and stability of the selected features. Follow-up angiography showed ISR in 30 patients. Carotid plaque length and thickness were independently associated with ISR (multivariate analysis); regarding the conventional model, the area under the curve (AUC) was 0.84 and 0.82 in the training and validation cohorts, respectively. The corresponding AUC values for the radiomics-based model were 0.87 and 0.82, and those for the optimal combined model were 0.88 and 0.83. Plaque length and thickness could independently predict post-CAS ISR, and the combination of radiomics and plaque features afforded the best predictive performance.

## 1. Introduction

Atherosclerotic stenosis of the carotid artery causes 10–15% of ischemic strokes [1]. Carotid artery stenting (CAS) is one of the standard treatment strategies for the management of symptomatic carotid stenosis and has emerged as an alternative to carotid endarterectomy (CEA) [2]. However, after follow-up for 6 months to 2 years, the incidence of in-stent restenosis (ISR) has been found to range from 3.3% to 21% [3,4,5]. The presence of ISR may act as a key factor influencing the long-term benefit and safety profile of CAS [6,7].

Currently, factors that predispose individuals to ISR can be classified as patient-related, procedure-related, and lesion-related. A previous study reported that an increase in C-reactive protein levels 48 h after CAS is a predictor of the occurrence of ISR at 6 months [8]. However, a preoperative assessment of lesion-related factors is necessary for the development of preventive treatment strategies.

Plaque characteristics assessed by carotid computed tomography angiography (CTA), such as plaque length [9], plaque texture [10], plaque thickness [11], plaque ulceration [12], and plaque enhancement [13], are strongly associated with the development of ischemic stroke. Accurate identification of these features can help in risk stratification among patients with carotid atherosclerotic stenosis. However, whether these carotid plaque characteristics are associated with carotid stent restenosis and whether these characteristics are effective in predicting ISR needs to be explored further.

High spatial resolution magnetic resonance imaging (MRI) can detect changes associated with increased carotid plaque vulnerability, such as intraplaque hemorrhage, lipid-rich necrotic cores, and thinning/rupture of the fibrous cap. However, multisequence carotid plaque MRI is cost- and time-intensive. In contrast, carotid computed tomography angiography (CTA) is a fast, simple, and widespread method that has become an established noninvasive phenotyping tool for patients with suspected carotid stenosis, providing an alternative for patients without access to MRI. The method affords the visualization and quantitative assessment of the degree of stenosis and plaque morphology [14]. However, it is difficult to visually distinguish between fibrous, lipid, and intraplaque hemorrhagic features due to the overlap of HUs. Machine learning algorithms have recently emerged as new assessment tools capable of sophisticated evaluations of radiological features extracted by CTA [15] and have been well utilized in risk assessment and personalized treatment decision-making in patients with coronary plaques [16]. Therefore, in this study, we aimed to screen preoperative clinical information, carotid plaque characteristics determined using CTA, and radiomics characteristics that could effectively predict the development of ISR after CAS and that could be combined to maximize predictive efficacy.

## 2. Materials and Methods

### 2.1. Study Design and Participants

This was a single-center retrospective database analysis of patients who underwent CAS. Potential subject records were screened via a review of consecutive CTA carotid examinations performed from April 2015 to March 2020. Patients who met the following criteria were included: (1) presence of 50% symptomatic stenosis or 70% asymptomatic stenosis in the extracranial carotid artery and receipt of successful CAS at our institution and (2) availability of complete data on clinical risk factors in medical records and follow-up records. The exclusion criteria were as follows: (1) a lack of follow-up data; (2) presence of nonatherosclerotic arterial stenosis (such as cerebral arteritis); (3) serious illnesses such as severe heart, lung, kidney disease or malignancy that are likely to cause death within the 6-month follow-up period; and (4) poor image quality, preventing the assessment of plaques and extraction of histological features. The protocol for this retrospective data analysis was approved by the institutional review board of JinLing Hospital, and the requirement for patients’ informed consent was waived.

### 2.2. CTA Technique

Carotid CTA examinations were performed using a second-generation dual-source CT scanner (SOMATOM Definition Flash, Siemens Healthcare, Forchheim, Germany) by using the following parameters: detector configuration, 64 × 0.6 mm; tube potential, 100 kV; tube current, 120 mAs; collimation, 128 × 0.625; pitch, 1.2; gantry rotation time, 0.33 s; field of view, 28 cm; slice thickness, 1 mm. Image reconstructions were rendered using the B20 intermediate reconstruction algorithm with the following characteristics: field of view, 160 mm; matrix size, 512 × 512; slice thickness, 1 mm; and interval, 1 mm. A bolus of 60 mL of high-iodine concentration contrast medium (400 mgI/mL; Iomeron 400, Bracco, Milan, Italy) was injected through an 18G cannula inserted into a superficial vein of the antecubital fossa of the right arm by using an automatic dual-head injector (Medrad Stellant Dual, Medrad, Palo Alto, PA, USA) at the rate of 5 mL/s followed by a 40-mL saline bolus injected at the rate of 4 mL/s. A bolus-tracking technique was used to trigger the acquisition at 5 s after a threshold of 100 HU was reached in the aortic arch. The scans were performed from the aortic root to 3 cm above the skull before and after contrast injection. All CTA images were reconstructed in axial, coronal, and sagittal orientations.

### 2.3. CAS

All procedures were performed via the femoral approach by using the Seldinger method. After the target stenosis was confirmed by angiography, a 6F or an 8F guiding catheter was navigated to the common carotid artery proximal to the stenosis. After a distal embolic protective device was released, a self-expandable stent, such as a closed-cell stent (Wallstent; Boston Scientific, Natick, MA, USA) or an open-cell stent (Precise; Cordis, Miami Lakes, FL, USA or Acculink; Abbott Vascular, Redwood City, CA, USA) of suitable size was then located and deployed. Pre-dilation was performed in some patients with severe stenoses. Complete angiography was performed to identify residual stenosis immediately after stenting. After the procedure, all patients were on long-term antiplatelet therapy.

### 2.4. Clinical and Imaging Data

After a thorough examination of the patients’ electronic medical records, two investigators recorded age; sex; clinical data including the presence of hypertension (blood pressure of >140/90 mmHg or use of antihypertensive medication), diabetes (hemoglobin A1c level of ≥6.5% or use of diabetes medication), and smoking history; any history of cardiovascular and old infarction events; and the type of stent (open-cell/closed-cell stent), with or without predilatation and residual stenosis. Laboratory data included total cholesterol, low-density lipoprotein, high-density lipoprotein, homocysteine, C-reactive protein, and hemoglobin A1c levels; white blood cell count; neutrophil and lymphocyte percentages; and mean platelet volume.

### 2.5. Conventional CTA Plaque Analysis

All CTA images were sent to a dedicated workstation (Syngo.Via, Siemens, München, Germany) for image analysis. To minimize potential variations due to inter-reader variation, all CTA findings were assessed by consensus by two neuroradiologists (XQC and CSZ) with 10 years of experience in the interpretation of cardiovascular CT results. (1) The degree of stenosis was assessed using the North American Symptomatic Carotid Endarterectomy Trial (NASCET) method [17]. (2) Plaque type was determined by obtaining three tissue density measurements from the area of least visual density at the level of maximum stenosis of the plaque. Measurements were obtained from a circular area of 2 mm^2^, and the smallest value was recorded. The texture was defined as soft when the median attenuation was approximately 40–50 HU and calcified when the increase in attenuation was >130 HU, according to previous literature [18]. (3) Plaque thickness was determined (linear measurement of the maximum axial dimension of the plaque on a single axial slice with the largest plaque thickness). (4) Plaque ulceration was defined as extension of contrast material beyond the vascular lumen of the plaque, usually by at least 1 mm [19]. (5) Plaque enhancement was defined as enhancement of ≥10 HU after contrast administration [18]. (6) Plaque length (in mm) was defined as the distance from the proximal to the distal shoulder of the plaque. (7) Positive remodeling was defined as an outer vessel diameter 10% greater than the mean of the diameter of the segments immediately proximal and distal to the plaque [20].

### 2.6. Radiomic Calculations

All CTA datasets in the Digital Imaging and Communications in Medicine (DICOM) format of the identified subjects were exported to an encrypted external hard drive. Radiomics feature extraction and dimensionality reduction were performed using the Deepwise research platform (Deepwise Inc., Beijing, China, http://label.deepwise.com (V1.6.2)). An axial section that maximized the inclusion of plaque features was selected, and an experienced radiologist (XQC with 10 years of experience in neuroimaging) manually delineated the region of interest (ROI) and covered as much of the plaque as possible. To assess the reproducibility of the CTA-based radiomics analysis, after 2 weeks, 30 patients in the development set were selected randomly, and their plaques were resegmented by another radiologist (Z.D., 3 years of experience in neuroimaging) to evaluate interobserver reproducibility by calculating intraclass coefficients (ICC).

Regarding logistic regression, the backward likelihood ratio elimination method was used to determine the most important and independent discriminating features. The features were weighted by their corresponding logistic regression coefficient. The best hyperparameters of each model were determined based on five-fold cross-validation. Radiomics signatures were verified in the validation cohorts. Figure 1 shows the radiomics workflow of this study.

### 2.7. Follow-Up Assessments

All patients underwent follow-up assessments with duplex ultrasound or CTA at the hospital outpatient clinic at 3, 6, and 12 months after the CSA procedure, and those suspected of having ISR underwent further digital subtraction angiography (DSA). ISR was determined by DSA to be ≥50% stenosis within the treated vessel, according to NASCET criteria.

### 2.8. Statistical Analysis

Statistical analysis was performed using dedicated statistical software (SPSS 25.0 statistical package; SPSS Inc., Armonk, NY, USA: IBM Corp.) and the Deepwise research platform (https://keyan.deepwise.com, V1.6.2). Categorical variables are reported as percentages, and continuous variables as means (SDs) or medians and interquartile ranges (IQRs). Student’s *t*-test and the non-parametric Mann–Whitney test were used to assess if there were significant differences between patients with and without ISR. Interobserver agreements were calculated by ICC. Features with ICC > 0.75 were considered to be consistent. Cox regression was performed to determine which baseline characteristics were independently associated with ISR. Receiver operating characteristic (ROC) curves were constructed, and ISR prediction models were developed using clinical, conventional plaque features and selected radiomics features to calculate the area under the ROC curve (AUC) and sensitivity and specificity. The DeLong test was used to compare the radiomics model with the traditional and combined models.

## 3. Results

### 3.1. Characteristics of the Study Participants

A total of 230 patients who underwent carotid CTA and were successfully treated with CAS were initially reviewed. We excluded four patients without a six-month postoperative follow-up record, three patients with poor-quality CTA images, and two patients with a diagnosis of vascular entrapment. Finally, 221 patients (186 men and 35 women) were analyzed in this study. The follow-up time ranged from 6 to 72 months. Follow-up angiography demonstrated ISR in 30 patients (13.6%). The average time for ISR development was 16.8 months. Demographic and clinical characteristics of the patients with or without ISR are shown in Table 1.

The ISR (+) and ISR (−) groups showed significant differences in past cerebral infarction (*p* = 0.022), degree of carotid stenosis (*p* = 0.009), plaque type (*p* = 0.033), plaque length (*p* < 0.001), plaque thickness (*p* < 0.001), and positive remodeling (*p* = 0.004), which are plaque characteristics measured by CTA. Laboratory tests for mean platelet volume (*p* = 0.016) and homocysteine (*p* = 0.035) showed significant differences between the two groups (Table 1).

### 3.2. Traditional Predictive Models

Multivariate Cox regression analysis showed that plaque length (HR 1.12, 95% CI 1.06–1.18, *p* < 0.005) and plaque thickness (HR 1.79, 95% CI 1.26–2.55, *p* < 0.005) were significantly associated with ISR (Table 1).

Combining image feature-based carotid plaque length and plaque thickness to build a traditional model to predict ISR yielded an AUC of 0.84 (95% CI: 0.77–0.91) for the training cohort with a sensitivity of 77% and specificity of 72%, and an AUC of 0.82 (95% CI: 0.73–0.90) for the validation cohorts, with a sensitivity of 73% and specificity of 71% (Figure 2A,B).

### 3.3. Radiomics Prediction Model

With the exception of three radiomics features with ICC values below 0.75, the interobserver reproducibility for measuring radiomics features was good, with ICC values ranging from 0.81 to 1.

A total of 2107 radiomics features were automatically computed from the pre-processed images, including the original image and corresponding wavelet images. The extracted features can be categorized into seven groups: NGTDM Features (Neighbouring Gray Tone Difference Matrix), GLDM Features (Gray Level Dependence Matrix), Gray Level Run Length Matrix (GLRLM Features), Gray Level Size Zone Matrix (GLSZM Features), Gray Level Co-occurrence Matrix (GLCM Features), Shape Features, and First-order Features. Then, we constructed a classifier to distinguish ISR. First, the computed features were normalized with zero-mean and unit variance. Second, the least absolute shrinkage and selection operator (LASSO) was used for feature selection. Third, we used logistic regression analysis to predict ISR.

Five-fold cross-validation was used to simulate the discriminatory power of the radiomics and conventional parameters. The radiomics model has shown good performance in predicting IRS (AUC: 0.87 [95% CI: 0.81–0.93] and 0.82 [95% CI: 0.74–0.90] in the training and validation cohorts, respectively; *p* < 0.001; Table 2, Figure 2A,B).

### 3.4. Combined Prediction Model

Combining radiomics and plaque features to create a combined prediction model yielded slightly higher performance than the conventional and radiomics models, with an AUC of 0.88 (95% CI: 0.82–0.95) for the training cohort and 0.83 (95% CI: 0.74–0.91) for the validation cohort (Table 2, Figure 2). The DeLong test revealed no significant differences among the conventional, radiomics, and combined prediction models in the training and validation cohorts (all *p* > 0.05).

By integrating radiomics features and carotid plaque features, a radiomics-based nomogram was created in the training cohort, as shown in Figure 3. In the nomogram, plaque length showed the highest weight, indicating that it was the most important predictor of ISR.

## 4. Discussion

This study evaluated preoperative clinical variables, laboratory indicators, CTA plaque features, and radiomics-related features to predict ISR after CAS. The results demonstrate that no laboratory indicators were effective in predicting ISR in this study; plaque length and thickness are significantly associated with ISR and are independent predictors of restenosis; and the conventional prediction model that combined the two had good predictive efficacy. In addition, the radiomics-based and combined models also exhibited good ability in predicting ISR and have been cross-validated with good stability.

ISR is associated with recurrent cerebrovascular events and is an important factor endangering the long-term safety and efficacy of CAS [6,7]. Previous studies have used laboratory indicators to predict the development of ISR, such as mean platelet volume, which has been shown to be associated with the development of IRS after CAS [21], but another study came to the opposite conclusion [22], and no laboratory indicators were found to distinguish or predict ISR, possibly due to the instability of hematological indicators, which are influenced by drugs as well as ethnic specificity. Another study showed that the preoperative neutrophil-to-lymphocyte ratio was a predictor of ISR in CAS patients, but the AUC was 0.58, which was not a strong predictor [23]. Our results show that no laboratory parameters were independently associated with ISR; therefore, this study concluded that the use of laboratory indicators to predict ISR is not sufficiently reliable.

In addition to laboratory parameters, this study assessed the correlation between CTA plaque characteristics and ISR and showed that both plaque length and thickness were independent predictors of ISR. Moreover, in the nomogram, the plaque length demonstrated the highest weight, indicating it was the most important predictive factor for ISR. Plaque length has been used as a risk factor in coronary artery stenting for patients with coronary artery disease [24,25], and lesion length greater than 20 mm is the strongest determinant of functional repercussions in the coronary artery [25]. For intracranial atherosclerosis stenting, a longer stenosis length has been reported to be an independent risk factor for ISR [26]. Subsequently, carotid plaque length was studied as an independent predictor of carotid plaque leading to microembolic signals [9] and ipsilateral carotid blood flow compromise [27] and was found to be independently associated with perioperative stroke and death in patients treated with CEA and CAS [28]. In addition, carotid soft plaque thickness is associated with plaque vulnerability, with each 1-mm increase in plaque thickness associated with a 2.7-fold increase in the odds of having a stroke or TIA [29]. In the present study, the AUC obtained after combining plaque length and plaque thickness to predict ISR was greater than 0.8, indicating that these two morphological variables play important roles in predicting IRS.

The results of this study agree with the previous findings reported by Bonati et al. [28], in which increased stenosis length at baseline independently predicted recurrent carotid stenosis after endovascular treatment, leading to the interpretation that endovascular treatment of longer lesions resulted in elastic retraction of the arterial wall and injury of the intima as well as more pronounced neointimal hyperplasia. In addition, this study speculates that the stent placed in CAS is made with a special type of metal with certain fatigue properties, which may lead to fatigue and deformation after prolonged asymmetric plaque compression or excessive plaque length and may be a potential cause of postoperative restenosis in patients. Moreover, carotid CTA allows easy measurement of morphological parameters such as carotid plaque length and plaque thickness, which will facilitate targeted treatment selection and follow-up strategies.

In comparison with visual observation or CT threshold calculations, plaque radiomics allows for a more sophisticated evaluation of imaging features based on automated extraction of multiple forms of imaging data that cannot easily be perceived by the human eye, further improving risk prediction for patients by integrating imaging, clinical, and bioinformatic phenotypic features. In this study, we used multivariate logistic regression analysis of 2107 carotid CTA plaque-based radiomics features to select the five most important predictors that showed a satisfactory predictive efficacy, with the AUCs for the training and validation sets being 0.87 and 0.82, respectively. A recent study used ultrasound (US)-based radiomics and clinical features to build a nomogram for identifying symptomatic carotid plaques with high diagnostic performance, with AUCs above 0.9 for both the training and validation sets [29]. Zhang et al. [30] used a histological model of plaque extracted using multiparametric MRI to accurately distinguish between symptomatic and asymptomatic carotid plaques, with AUCs above 0.9 in both the validation and test sets, and sensitivities and specificities above 90%. In addition, related studies have used CTA to extract plaque texture features from patients with carotid atherosclerosis, which could help identify plaques that may become culprit lesions and thereby predict future acute ischemic events [31,32]. We developed and validated an integrated model combining radiomics features with significant plaque predictors, which showed improved ability for predicting IRS. Cross-validation simulations indicate that our results are robust.

This study has some limitations. First, this was a retrospective study with a small sample size; therefore, we did not employ an independent internal validation set. We calculated all diagnostic scores using five-fold cross-validation, which yielded a robust estimate of diagnostic accuracy. Second, our results were obtained from a single center, using the same scanner and reconstruction settings, which limited the generalizability of our findings. The third limitation was the lack of semiautomatic segmentation software for carotid CTA, which prevented the analysis of more of the plaque components. The fourth limitation was the need for manual segmentation of plaques before radiomics analysis. To address variability in plaque segmentation between experts, a second neuroradiology fellow manually contoured plaques using the same processing pipeline. Fifth, the combined model of radiomics and plaque features predicted the AUC of ISR better than the radiomics and conventional plaque models, but the results showed no statistical difference after the DeLong test. This may be attributed to the fact that the extraction of radiomics features in this study only segmented the dimension of the largest plaque region and did not use a three-dimensional approach to segmentation, which may have lost some powerful radiomics features. Therefore, we plan to use a three-dimensional segmentation approach to extract radiomics information in our follow-up study, which may improve ISR prediction performance in the future.

In summary, our study demonstrates that plaque characteristics such as lesion length and thickness measured by carotid CTA have independent predictive value for ISR after CAS and that these features are easily accessible and suitable for application in clinical practice. In contrast, the radiomics and combined models also provide stable and good predictive efficacy for ISR, and there is hope for further improving the predictive efficacy of ISR patients using three-dimensional extraction of radiomics features and algorithm enhancements in subsequent studies.

## Figures and Tables

**Figure 1 jcm-11-03234-f001:**
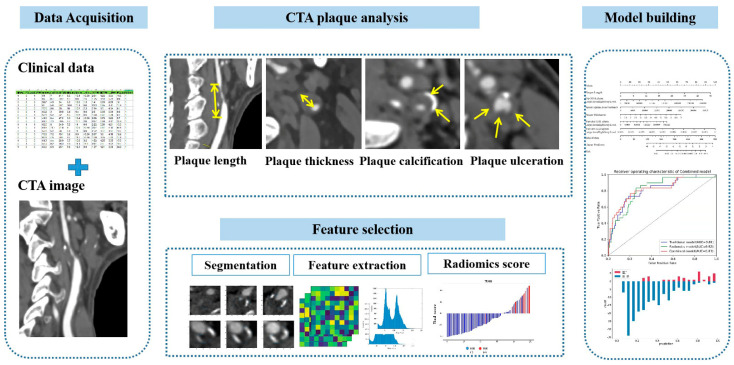
A flowchart of model development. Clinical, laboratory, and imaging data were collected and plaque analysis was performed on carotid CTA. Manual segmentation was performed using an axial slice that best represents the plaque features. The backward likelihood ratio elimination method was used to determine the most important and independent discriminating features. Selected radiomics features were used to build radiomics models. The model was evaluated using the area under the receiver operating feature curve for the training and validation cohorts. The predictive power of the classifier was assessed by 5-fold cross-validation.

**Figure 2 jcm-11-03234-f002:**
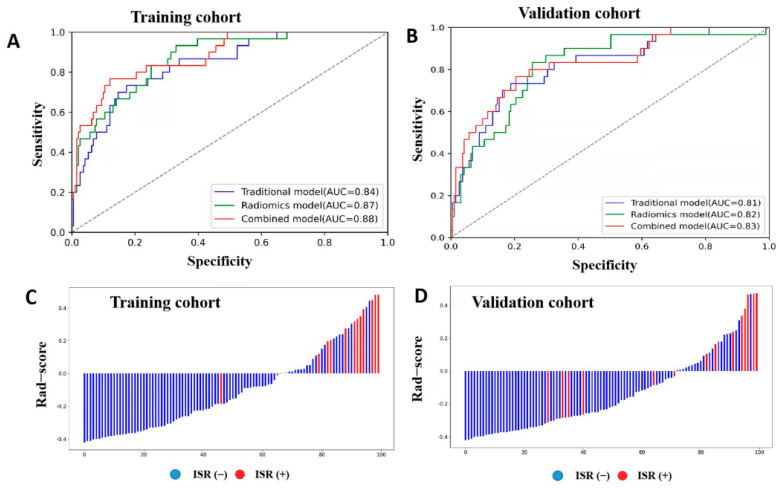
Receiver operating characteristic (ROC) curves and radiomics score. ROC curves for the performance of the three prediction models (traditional, radiomics, combined) in the training (**A**) and validation cohorts (**B**). Radiomics score (Rad-score) of each patient in the training (**C**) and validation cohorts (**D**) showed the association of high Rad-score with risk of in-stent restenosis. Red represents patients with in-stent restenosis, whereas blue represents patients without in-stent restenosis.

**Figure 3 jcm-11-03234-f003:**
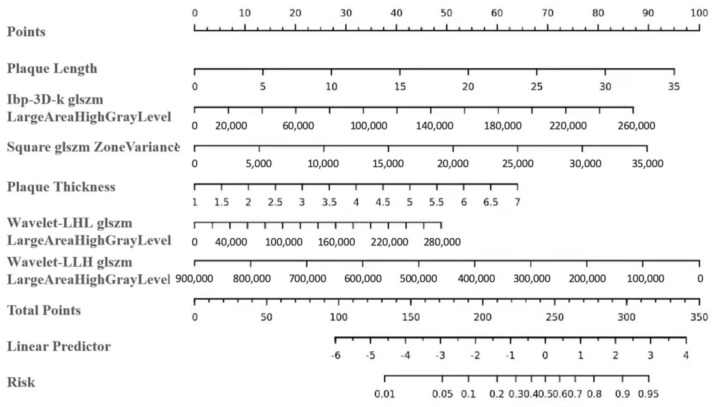
A radiomics-based nomogram incorporating plaque length, plaque thickness, and radiomics features for identifying carotid in-stent restenosis was developed in the training dataset. The probability score for stent restenosis is labelled on each axis, increasing from left to right. To use the nomogram, find the position of each variable on the corresponding axis, draw a vertical line on the point count axis to indicate the number of points, add up the number of points for all variables and draw a line on the total point count axis to determine the probability of in-stent restenosis on the lower line of the nomogram.

**Table 1 jcm-11-03234-t001:** Clinical, laboratory, and carotid artery plaque characteristics of the patients.

Characteristic	All Patients (*n* = 221)	ISR Group (*n* = 30)	Non-ISR Group (*n* = 191)	Univariate Analysis *p*-Value	Multivariate	
					HR (95% CI)	*p* Value
Demographics						
Age, y, mean ± SD	66.89 ± 8.07	68.73 ± 6.34	66.60 ± 8.29	0.37		
Male, *n* (%)	186 (84.16)	28 (93.33)	158 (82.72)	0.23		
Risk factors						
Smoking, *n* (%)	100 (45.25)	13 (43.33)	87 (45.55)	0.82		
Hypertension, *n* (%)	195 (88.24)	29 (96.67)	166 (86.91)	0.22		
Diabetes mellitus, *n* (%)	78 (35.29)	10 (33.33)	68 (35.79)	0.79		
Coronary artery disease, *n* (%)	50 (22.62)	9 (30.00)	41 (21.47)	0.30		
Past cerebral infarction, *n* (%)	77 (34.84)	16 (53.33)	61 (31.94)	0.02	0.59 (0.26–1.32)	0.20
Laboratory parameters						
Neutrophil percentage, %, mean ± SD	61.98 ± 8.93	62.62 ± 8.95	61.88 ± 8.94	0.59		
White blood cell count, 10^9^/L, mean ± SD	6.99 ± 2.38	7.97 ± 3.61	6.84 ± 2.10	0.17		
Lymphocyte percentage, %, mean ± SD	27.80 ± 8.22	27.39 ± 7.64	27.87 ± 8.32	0.77		
Glycated hemoglobin, %, mean ± SD	6.47 ± 1.20	6.30 ± 1.12	6.46 ± 1.20	0.43		
Mean platelet volume, fL, mean ± SD	10.82 ± 1.25	11.29 ± 1.00	10.82 ± 1.25	0.02	1.32 (0.94–1.83)	0.11
C-reactive protein level, mg/L, mean ± SD	4.10 ± 6.97	3.76 ± 4.86	4.10 ± 6.97	0.59		
Low-density lipoprotein, mmol/L, mean ± SD	2.43 ± 0.90	2.36 ± 0.83	2.44 ± 0.91	0.86		
High-density lipoprotein, mmol/L, mean ± SD	1.04 ± 0.34	0.99 ± 0.18	1.05 ± 0.35	0.57		
Total cholesterol, mmol/L, mean ± SD	4.14 ± 1.09	4.14 ± 0.85	4.14 ± 1.12.	0.74		
Homocysteine, mmol/L, mean ± SD	16.06 ± 9.15	19.66 ± 12.00	15.49 ± 8.52	0.04	1.02 (0.98–1.07)	0.28
Carotid artery stenting						
Open cell stent, *n* (%)	158 (71.49)	23 (76.67)	135 (70.68)	0.50		
Pre-dilation, *n* (%)	192 (86.88)	26 (86.67)	166 (86.91)	0.31		
Residual stenosis, %, mean ± SD	10.38 ± 11.27	13.83 ± 14.37	9.84 ± 10.65	0.18		
Lesions						
Stenosis, %, mean ± SD	77.03 ± 13.38	83.10 ± 10.76	76.07 ± 13.52	0.01	2.17 (0.06–73.87)	0.67
Soft plaques *n* (%)	164 (74.21)	27 (90.00)	137 (71.73)	0.03	3.24 (0.98–10.67)	0.05
Lesion length, mm, mean ± SD	15.49 ± 7.00	24.50 ± 5.60	14.08 ± 6.09	<0.001	1.12 (1.06–1.18)	<0.005
Plaque thickness, mm, mean ± SD	3.30 ± 1.26	4.51 ± 1.09	3.10 ± 1.18	<0.001	1.79 (1.26–2.55)	<0.005
Plaque ulceration, *n* (%)	83 (37.56)	11 (36.67)	72 (44.27)	0.91		
Plaque enhancement, *n* (%)	81 (36.65)	13 (43.33)	68 (35.60)	0.41		
Positive remodeling, *n* (%)	101 (45.70)	21 (70.00)	80 (41.88)	0.004	0.65 (0.26–1.61)	0.35

Abbreviations: ISR in-stent restenosis. Data are displayed as mean (SD) or number (percent).

**Table 2 jcm-11-03234-t002:** Performance of the models in training and validation cohorts.

Predictive Models	Cohort	AUC (95% CI)	Sensitivity	Specificity
Traditional model	Training	0.84 (0.77–0.91)	0.77	0.72
	Validation	0.81 (0.73–0.90)	0.73	0.71
Radiomics model	Training	0.87 (0.81–0.93)	0.77	0.75
	Validation	0.82 (0.74–0.90)	0.77	0.74
Combined model	Training	0.88 (0.82–0.95)	0.80	0.79
	Validation	0.83 (0.74–0.91)	0.77	0.76

AUC, area under the curve.

## Data Availability

Not applicable.

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
