# Peer review of "Prediction of Carotid In-Stent Restenosis by Computed Tomography Angiography Carotid Plaque-Based Radiomics"

_jcm, 2022, doi:10.3390/jcm11113234_

Round 1

Reviewer 1 Report

This is an interesting retrospective study that describes how pre-clinical characteristics and pre-CAS CTA imaging data may help predict intra-stent re-stenosis (ISR) after CAS. 

The work is original and addresses a gap in literature. Methods are sound and figures and table clear and efficient. 

Methods:

  1. please define what was considered "severe heart, lung, or kidney disease" as exclusion criteria
  2. please define what was considered "poor quality" images as exclusion criteria 
  3. please report the number of patients that received dilation before stent deployment 
  4. please define what was considered cardiovascular risk factor
  5. the introduction briefly mentioned how CRP levels after CAS may predict ISR, why CRP was not collected in this study?

Results:

  1. please report the average time of when ISR developed.

Discussion:

  1. as you mentioned there are no pre-CAS labs that predicts ISR, it may be interesting reporting CRP levels given that was previously associated with ISR
  2. The limitations mentioned are important, especially the retrospective nature of the study and the small sample size 
  3. the effort of the authors to use a 3D approach to better characterize the plaque is interesting and appreciated 

Reviewer 2 Report

Summary:

Cheng and colleagues have used pre-operative computed tomography angiography (CTA) of patients with carotid stenosis treated with carotid artery stenting (CAS) to predict in-stent restenosis (ISR). They used carotid plaque characteristics from traditional reading and image analysing tools but also added information from more advanced image analysis/radiomics. The aim of the study was to explore how CT-based carotid radiomic features can add to the predictive value of classical CT parameters.

They find that plaque length and thickness independently could predict post-CAS ISR. The combined model of radiomics and plaque features predicted the AUC of ISR better than the radiomics and conventional plaque models, but the DeLong test revealed no significant differences among the conventional, radiomics and combined prediction models in the training and validation cohorts (all P > 0.05).

The paper addresses an important issue in the discussion about what is the optimal treatment for carotid stenosis with the aim of preventing cerebrovascular events. The study is large, 221 patients were analysed. The study has an advantage for CT analysis of plaque components since both pre- and post contrast images were acquired.  

General comments:

  • The study does not use the acquired CT data sufficiently. The plaque component parameters obtained from the software (Coronary Plaque Analysis) are too few. A much more detailed plaque analysis with more voxel statistics including more HU categories, volume and wider measurements before and after contrast must be done. Before introducing a more advanced imaging parameter/radiomics – the most information should be extracted with already known methods.
  • The article does not clearly define the difference between coronary and carotid plaques when it comes to choices of methods and references, these are also mixed up and used incorrect. The authors must eliminate the ambiguity throughout the paper. Coronary and carotid atherosclerosis have a lot in common, but several differences – and the imaging possibilities are different, at least due to size and movements.
  • The choice of only one axial slice 1 mm thick covering the plaque for radiomics analysis gives very little information. It is not sufficient for detailed plaque characteristics – and the whole point of extracting more data from the images / radiomics is not fulfilled. The data included in this study is not sufficient to evaluate what the potential for a radiomics analysis can be in predicting ISR. The authors admit this limitation in the discussion line 366 “This may be attributed to the fact that the extraction of radiomics features in this study only segmented the dimension of 367 the largest plaque region and did not use a three-dimensional approach to segmentation, 368 which may have lost some powerful radiomics features.“
  • Figure 1 is not readable due to very poor resolution, and can not be evaluated.
  • Radiation protection – why were the patients scanned from aortic root to above the skull pre- contrast as only the stenosis to be stented was of interest for the HU measurements?

Specific comments:

  • Introduction:
    • Line 49-50 “the high volume of the plaque components with 49 radiodensities < 0 Hounsfield units (HU) [11], and high-grade calcification [12]”, reference 11 and 12 refers to atherosclerotic carotid stenosis and NOT coronary … as stated in the paper, please rewrite line 50-51: “However, the prediction of postoperative ISR development after CAS based on a 52 preoperative assessment of carotid plaques has not been well-researched” - and find correct references for statements. Findings from the carotid artery should be reported specifically as that is the theme of this study.
  • Materials and Methods,
    • 2 CTA technique page 2/12
      • Line 91 please specify/add reconstruction method and reconstructed voxel size. It would be clarifying to specify CT acquisition- and reconstruction parameters separately.
      • The amount of contrast media administered is not described.
  • 5 Conventional CTA plaque analysis
    • Line 124 -126, the software is produced for coronary, – not carotid, plaque analysis, please specify this – and eventually debate pitfalls when using the software for carotid.
    • Line 127-132. The text is unclear – is more than one plaque chosen for analysis?
    • Line 137/138 – how are the plaques classified, mean of all voxels – please explain. Why is the interval HU 40-50 chosen?
    • Line 138 – plaque thickness – lumen decides, not plaque – this could miss the thickest part due to arterial remodelling.
    • Line 140, Please explain or show by image what is meant.
    • Line 140, outward remodelling – what if the plaque is in the bulbus/wide area of the bifurcation?
  • 6 Radiomic calculations
    • Line 152/153 – is it correct that only one axial slice from the plaque is used in the radiomic calculations? Describe what was delineated in the ROI ?
    • Line 156 – reader two resegmented – what does that mean? segmentation is not described above.
  • Table 1
    • Plaque type should be 2 categories, soft or calcified, not specified. Plaque surface morphology - 2 categories, not specified.

Round 2

Reviewer 2 Report

Most important:
1) The authors have not provided more detailed plaque analysis with more voxel statistics as I asked for. Before introducing an advanced imaging parameter/radiomics – the most information should be extracted with "state of the art" from already known methods. Since this papers aim is to explore radiomics models - that is basically a more advanced voxel analysis - compared to traditionally availabla data, the latter must be as good as possible. The authors have had softvare available for volume measurements of HU categories - but did not use it.

2)The methods for plaque CTA HU analysis on voxel basis is now completely changed, in the first version the authors described that they used the software "Coronary Plaque Analysis, Syngo.Via FRONTIER, version 5.0.0, Siemens Healthcare"(former line 124) for this. In the second version the use of this software is omitted from the manuscript and a manual method for ROI placement and analysis is added. What method did they really use? Their new description of manually methods chosen gives the paper a poorer quality since they are too simple and they have measured at different plaque locations, thus not comparing methods with the same input data.

3) The new version of the Introduction does not contain relevant references for this work. The Introduction is not able to explain to the reader what is the current knowledge, and what are the knowledge gaps for predicting ISR from CTA. The ISR literature is also lacking in their discussion.

The paper has the potential for improving if the data collected are used more wisely. The theme is highly relevant - and we need to know where to use radiomic techniques in radiology to better disease understanding and patient care.

Author Response

Reviewer2

Most important:
1) The authors have not provided more detailed plaque analysis with more voxel statistics as I asked for. Before introducing an advanced imaging parameter/radiomics – the most information should be extracted with "state of the art" from already known methods.

Since this papers aim is to explore radiomics models - that is basically a more advanced voxel analysis - compared to traditionally availabla data, the latter must be as good as possible. The authors have had softvare available for volume measurements of HU categories - but did not use it.

Responses: I am very sorry that my revision was not able to satisfy you. For carotid CTA plaque analysis, there are real difficulties in differentiating plaques in more detail by physician delineation of ROI without software. SIEMENS offers software for coronary plaque analysis, but as you said before, there are differences in the analysis of carotid plaque using this software. Carotid CTA images can be imported into the coronary analysis software and although the volumetric measurements are displayed, these data can only be used for reference and may result in inaccurate results if used for analysis. If we had specific software for carotid plaque analysis, we would certainly use it and be able to help with the more detailed plaque analysis you mentioned.

In addition, we agree with you that radiomics models are currently outperforming conventional models in most studies. The similarity in predictive efficacy between the radiomics and traditional model in this study is a shortcoming of this study, and I have added a paragraph to the discussion section in the hope that the subsequent increase in sample size and the use of three-dimensional segmented plaques will further improve the radiomics models predictive efficacy.

2) The methods for plaque CTA HU analysis on voxel basis is now completely changed, in the first version the authors described that they used the software "Coronary Plaque Analysis, Syngo.Via FRONTIER, version 5.0.0, Siemens Healthcare"(former line 124) for this. In the second version the use of this software is omitted from the manuscript and a manual method for ROI placement and analysis is added. What method did they really use? Their new description of manually methods chosen gives the paper a poorer quality since they are too simple and they have measured at different plaque locations, thus not comparing methods with the same input data.

Responses: Thank you for your comments. We have consulted with SIEMENS regarding the software and the company replies that the coronary plaque analysis software cannot yet be used for parametric measurements of carotid plaque and that the carotid plaque composition and volume obtained by the software may be inaccurate. However, it can be used to indicate to the evaluators the presence and location of plaque and to facilitate subsequent manual measurements is feasible. We are concerned about misleading readers to use coronary software to measure carotid plaque and this is the main reason for our deletion of this paragraph.The actual measurement is still done by manually placing the ROI, which is the predominant method of measuring plaque composition in clinical practice. However, it is difficult to distinguish features such as fibrous, lipid and intraplaque haemorrhage due to the overlap of HU.

We appreciate and value your comments. The lack of software for carotid plaque analysis is a shortcoming that we are currently unable to address.Please consider giving this article a chance, it is very important for the postgraduate students who are participating in this study and will be graduating. We are already working on 3D plaque segmentation and hope that subsequent studies will further improve the predictive power of the radiomics models.

3) The new version of the Introduction does not contain relevant references for this work. The Introduction is not able to explain to the reader what is the current knowledge, and what are the knowledge gaps for predicting ISR from CTA. The ISR literature is also lacking in their discussion.

Responses: Thank you for your comments. There is a lack of articles on the use of carotid CTA plaques as well as radiomic features to predict ISR. Therefore, in the introduction section we focus on 1, the incidence of carotid ISR and the resulting poor prognosis; 2, laboratory studies and pitfalls in predicting ISR; 3, the relationship between carotid high-risk plaque features and ischaemic stroke occurrence, but these plaque features are less frequently used to predict ISR; and 4, the pitfalls of carotid CTA in segmenting plaque features due to overlap of HU. Therefore, this study hypothesised further histological feature extraction of plaques to combine conventional CTA plaque features with plaque radiomic features to construct an optimal prediction model.

    Carotid CTA plaque prediction of ISR is not mentioned in the introduction due to the lack of relevant literature. In the discussion section we have analysed separately:1 the studies and shortcomings of laboratory indices for the prediction of ISR, and the plaque length and thickness obtained in this study as independent predictors of ISR, and the analysis of plaque length as a risk factor for restenosis after coronary stenting in the coronary arteries. Stenosis length is a risk factor for ISR after intracranial atherosclerosis stenting. Carotid plaque length is a risk factor for blood flow compromise and carotid plaque thickness increases the risk of stroke occurrence.However, there is a lack of literature related to carotid CTA plaque characterisation for predicting carotid ISR. 2 In the radiomics discussion section, we analyse a study in which MRI was used to assess carotid plaque radiomic features to differentiate between symptomatic and asymptomatic plaques in the carotid artery and another study in which CTA was used to extract carotid plaque texture features to identify potentially culprit plaques. There is a lack of literature on radiomic models to predict carotid ISR.

   This study is a preliminary study using traditional plaque and radiomic plaque characteristics of carotid CTA to predict ISR, and as you mentioned, there are many technical flaws. Our study is still undergoing continuous improvement. Thank you for your comments, which are very important to improve the quality of our subsequent research.
